# How Do Financial Development and Renewable Energy Affect Consumption-Based Carbon Emissions?

**Abraham Ayobamiji Awosusi [1], Tomiwa Sunday Adebayo [2], Husam Rjoub [3,4,5] and Wing-Keung Wong [6,7,8,\*]**

[1] Department of Economics, Faculty of Economics and Administrative Science, Near East University, North Cyprus, Mersin 10, Nicosia 99138, Turkey

[2] Department of Economics, Faculty of Economics and Administrative Science, Cyprus International University, Mersin 10, Nicosia 99040, Turkey

[3] Department of Accounting and Finance, College of Administrative Sciences and Informatics, Palestine Polytechnic University, Hebron City P.O. Box 198, West Bank, Palestine

[4] Department of Business Administration, Abdul Haris College of Administrative Sciences, Makassar 90244, Indonesia

[5] Department of Business Administration, Faculty of Management Sciences, ILMA University, Karachi 75190, Pakistan

[6] Department of Finance, Fintech & Blockchain Research Center, and Big Data Research Center, Asia University, Taichung City 41354, Taiwan

[7] Department of Medical Research, China Medical University, Taichung City 40447, Taiwan

[8] Department of Economics and Finance, The Hang Seng University of Hong Kong, Shatin 999077, Hong Kong

\* Correspondence: wong@asia.edu.tw

**Abstract:** This paper bridges the gap in the literature by employing the novel quantile-on-quantile (QQ) approach, the quantile regression approach, and the nonparametric Granger causality test in quantiles to assess the effect of international trade on consumption-based carbon emissions ($CCO_2e$) in Uruguay. Our study incorporates other drivers of $CCO_2$ emissions, such as financial development and renewable energy, into the model. We find that, in the majority of the quantiles, exports, financial development, and renewable energy exert a negative impact on $CCO_2e$, and the influence of imports on $CCO_2e$ is positive in all quantiles. Moreover, the quantile regression approach is used as a robustness test for the quantile-on-quantile approach. The causal interaction from the regressors to $CCO_2e$ is evaluated using the nonparametric Granger causality test in quantiles. The outcome of the nonparametric Granger causality test in quantiles suggests that imports, exports, renewable energy, and financial development can predict $CCO_2e$ at different quantiles. Based on these outcomes, we recommend that the financial sector must strengthen its focus on giving funding to enterprises that embrace environmentally friendly technologies and incentivize them to employ other energy-efficient technologies for manufacturing reasons, thereby preventing environmental deterioration.

**Keywords:** financial development; renewable energy; export and import; consumption-based carbon emissions

## 1. Introduction

The primary goal of the Paris Agreement is to enhance the global effort to address the threat of climate change by reducing the global average temperature increase to below 2 °C, beyond the level of pre-industry within this century, and implementing efforts toward mitigating the global temperature rise to 1.5 °C [1–6]. Environmental degradation is becoming one of the world's most significant and severe challenges in the contemporary period. Because of greenhouse gas (GHG) emissions, the Earth's average temperature has risen by 1.4 °F since 1880, causing severe threats to other species worldwide [7–11].

Carbon emissions ($CO_2$) account for over 75% of global GHG emissions as a by-product of anthropogenic climate change, and they cause extreme weather events, such as

droughts, floods, torrential rain, and heatwaves, which have transpired frequently in the last decade. These severe occurrences significantly impact people, animals, and the environment [5,6]. To address these significant concerns, the United Nations holds a conference in Glasgow (COP26) on climate change to contextualize environmental trends and suggest solutions. One of the critical achievements of COP26 includes finalizing the Paris Agreement rulebook. This set of rules shows how countries are held accountable for delivering on their climate action promises and self-set targets under their Nationally Determined Contributions (NDCs) [8].

Uruguay is already dealing with the consequences of climate warming. As a signatory to the Paris Agreement, Uruguay has a unique opportunity to seek international support in addressing its environmental concerns. Reducing carbon emissions without inhibiting economic expansion has been a complicated challenge [9,12–15]. Numerous studies were conducted to ascertain the underlying cause of environmental deterioration, such as [16], which asserted that lowering environmental quality poses a severe threat to human existence, whereas renewable energy, international trade, economic growth, and financial development are essential for comprehending the ecosystem. In addition, refs. [17,18] confirmed that renewable energy reduces carbon emissions.

Uruguay is an emerging nation in Latin America where environmental deterioration is exerting a significant impact, perhaps through changing climate patterns or an increase in catastrophic events. In the current period, environmental challenges will serve an essential role in nations' sociological and economic advancement, particularly emerging ones, in the near future. It is impossible to comprehend the magnitude of these changes, especially in Uruguay, where the economy is rapidly developing and is expected to persist in the immediate future. Uruguay maintains the high energy required to continue its fast industrial growth, and traditional energy sources are being used to meet the country's quickly expanding energy demand. According to [19], the conventional energy utilization resources emit $CO_2$, worsening environmental sustainability and impacting human life in Uruguay. According to [20], any change in the environment can have a positive or negative impact on all phases of life.

Besides the above, SDG 7 of the UN Post-2015 Objectives is among the 17 thematic targets that propose the use of cheap and clean energy. For success, adaptations and reliance on integrated sustainability activities are essential. Renewable energy is crucial for lowering emissions and also contributing to economic growth in this regard. Besides renewable energy, financial development is crucial for Uruguay's economic expansion because any well-organized financial sector will obtain economic stability. Although Uruguay has achieved significant development in terms of financial legislation and supervision, improving the financial system's soundness, the depth of its financial system remains relatively low compared to other nations in the area.

On the other hand, it might increase energy consumption, which has negative environmental repercussions that must not be overlooked. Few studies (such as [14,19,21]) concentrated on various environmental considerations in Uruguay; moreover, none of them probed into the impacts of imports, GDP, exports, renewable energy, and financial development on consumption-based emissions (CCO₂e) in Uruguay. The computation of CCO₂e has been proposed as an alternative accounting method for measuring global emissions from a country's usage of goods and services. This new method of evaluating carbon emissions is known as the carbon footprint. The embedded emissions from production, transportation, and the sale of products and services are used to evaluate it [22,23]. Under the CCO₂ emissions assessment, international trade is the most common avenue for shifting responsibility regarding carbon emissions from producers to consumers. It symbolizes a significant shift in the documentation of $CO_2$ emissions utilizing a more comprehensive accounting system.

With this rationale, this research intends to make the following contributions. First, it investigates a novel proxy for carbon emissions—in particular, for consumption-based carbon emissions that are adjusted to international trade—which is particularly relevant

to a nation such as Uruguay, where the import and export of goods and services amount to 21.46% and 26.39% of the GDP, respectively, as of 2018. Second, for the model of the $CCO_2e$, this current study incorporates financial development and renewable energy use as the determinants of $CCO_2e$, as well as adding existing drivers such as international trade, which is measured by splitting it into imports and exports. Third, we utilize the quantile-on-quantile (QQ) method to investigate the impacts of imports, exports, the usage of renewable energy, and financial development on $CCO_2e$ in Uruguay. Moreover, [24] asserted that many econometric approaches are essential for producing biased research results, highlighting the need to advocate for using more sophisticated econometric approaches. The QQ technique has the benefit of accounting for the heterogeneity of the slope and being robust to outliers. Fourth, the causality-in-quantiles test detects the causality relationship between $CCO_2e$ and the regressors. The benefit of this procedure is to consider both the mean and variance of the causal linkage between the two parameters simultaneously. Currently, no study has been undertaken to establish the interconnection between financial development, imports, exports, and the usage of renewable energy in $CCO_2e$ in Uruguay utilizing the nonparametric causality-in-quantiles test and the QQ approach. Finally, this study examines Uruguay's uncertainties regarding anthropogenic global warming and its detrimental implications for the country's growth. It will also make policy suggestions regarding steps that policymakers could take toward achieving the desired green economy in the nearest future.

The next section presents a summary of studies, and Section 3 discusses related theoretical frameworks, data, and the methodologies used in our paper to show comprehensive analysis techniques. Section 4 presents the empirical analysis and the discussion, and Section 5 presents the conclusions and policy implications.

## 2. Literature Review

This portion of the article will discuss the prior studies on the connection between environmental degradation and imports, exports, renewable energy, and financial development.

### 2.1. Renewable Energy and Environmental Degradation

Significant studies have been conducted regarding the association between renewable energy consumption and $CCO_2e$ emissions. According to previous research, renewable energy sources are often regenerated and are highly acknowledged for their environmentally friendly capacity. Furthermore, due to an upsurge in energy demand, countries must investigate numerous strategies to encourage manufacturing and employ numerous types of renewable energy sources in the future. In the same respect, ref. [25] used G7 economies as an example to demonstrate the detrimental impact of renewable energy on consumption-based carbon emission ($CCO_2e$) for the period between 1990 and 2017. Furthermore, ref. [26] revealed the relevance of renewable energy in reducing $CCO_2e$ in Italy for the period between 1970Q1 and 2018Q4. Moreover, ref. [27] examined the link between renewable energy and $CCO_2e$ connection in Chile. The empirical research revealed that renewable energy is a critical resource for reducing $CCO_2e$ in Chile. Likewise, ref. [28] demonstrated the negative impact of renewable energy on $CCO_2e$ in the context of the Indian economy. Furthermore, ref. [29] used a case study of China and demonstrated that renewable energy is vital in reducing $CCO_2e$. As a result, the Chinese economy must encourage various renewable energy methods at the home and business levels. Moreover, ref. [30] revealed a negative effect of renewable energy on $CCO_2e$ in the Chilean economy. Moreover, ref. [31] investigated the renewable energy–$CCO_2e$ interconnection for Regional Comprehensive Economic Cooperation (RCEP). Meanwhile, the empirical analysis uncovered a negative relationship between $CCO_2e$ and renewable energy. In the context of the ten selected nations, the study of [32] identified a negative relationship between environmental degradation and renewable energy.

## 2.2. International Trade and Consumption-Based Carbon Emission

International trade is commonly acknowledged as a significant factor in $CO_2$ emissions [23,33]. Numerous studies on the interconnection between $CO_2$ emissions and international trade have split international trade into imports and exports to evaluate the influence of international trade on $CO_2$ emissions. Most of the previous research has assessed the relationship between $CO_2$ emissions and trade, while there are only a few investigations using $CCO_2$ emissions as a proxy for environmental degradation. For example, the research of [34] reported that imports decrease the quality of the environment while exports improve it in the case of nine exporting nations. However, for twenty Asian economies, the study of [22] established that exports decrease the level of $CCO_2e$ while imports increase $CCO_2e$, thereby offsetting each other. In the same context, ref. [25] concluded that imports have a positive relation with $CCO_2e$ and exports have a negative influence on $CCO_2e$ in G7 economies. Using the NARDL approach, the study of [33] revealed that positive (negative) shocks in exports tend to decrease (increase) the level of $CCO_2e$ n all MINT economies, while positive (negative) shocks in exports tend to increase (decrease) the level of $CCO_2e$ for the period ranging from 1990 to 2018. A panel analysis study conducted by [5] for the study of MINT nations revealed that exports decrease the level of $CCO_2e$, but imports increase $CCO_2e$. Moreover, ref. [35] probed into the association between international trade and $CCO_2e$ in Bolivia for the period ranging between 1970 and 2018. The empirical evidence shows that exports decrease the level of $CCO_2e$, but imports increase $CCO_2e$.

## 2.3. Financial Development and Environmental Degradation

A vibrant financial sector is critical for the development of both the economy and human beings. It is also critical to assess the environmental effect of financial development. There are studies showing the relationship between environmental quality and financial development; however, the results are conflicting due to different indicators used to measure financial development. Literature reveals that financial development could improve the environment's quality by lowering the environmental degradation level. For example, ref. [36] probed the effect of financial development on $CO_2$ in G8 nations. They concluded that financial development significantly decreases environmental degradation. Moreover, ref. [37] established a negative association between financial development and $CO_2$ in OECD economies. Regarding selected Asian economies, ref. [38] detected an adverse effect of financial development on $CO_2$. Moreover, ref. [39] assessed a similar connection for Malaysia. The empirical analysis indicates that financial development proves to be a roadblock to more environmental degradation practices. Moreover, ref. [40] established a negative connection between financial development and $CO_2$.

Furthermore, the line of literature has established a detrimental impact of financial development on the quality of the environment by increasing pollution. For example, ref. [41] assessed the influence of financial development on $CO_2$ in 15 selected Asian economies. In addition, the study observed that financial development had a positive influence on $CO_2$. In the case of the UAE, ref. [42] confirmed that financial development is a necessary component for increasing $CO_2$. Furthermore, ref. [43] conducted a study on South Asian economies for the period 1984 to 2015. The empirical evidence confirmed that financial development positively influences $CO_2$. The study of [44] documents a positive relationship between $CO_2$ and financial development in Latin American countries.

Prior literature examining the effect of imports, exports, renewable energy, and financial development on environmental degradation has been reviewed in this section. Although there have been several types of research on the subject, we could not uncover a detailed review on this connection for these indicators in Uruguay. As previously stated, no study for the case of Uruguay has employed consumption-based carbon emissions as an environmental degradation indicator. Furthermore, we were unable to discover any

research premised on the advanced econometric methodologies of the quantile-on-quantile approach and nonparametric causality approach for any economy in Latin American nations, which is part of the innovation of this current study. Evaluating the interrelationship with these procedures allows us to uncover the peculiarity of these approaches, culminating in accurate estimates that facilitate proactive policy choices.

## 3. Theoretical Framework, Data, and Methods

### 3.1. Theoretical Framework

This section describes the theoretical processes through which renewable energy, imports, exports, and financial development influence consumption-based carbon emissions (CCO₂e). According to [45], a change in inventories, resident spending abroad, formation of gross capital, the final domestic consumption demand from the government, and the household sector comprise CCO₂e. This trade-adjusted indicator encompasses the comprehensive carbon chain and helps determine carbon emissions generated in one country and absorption in other countries [27]. Therefore, we use imports and exports to measure the influence of international trade in this study.

As stated by [30], a rise in exports yields more items and services for destination countries to utilize while retaining less for domestic usage. Exports comprise goods and services produced in the country of origin and consumed in the recipient country. Therefore, CO₂ emissions from exports must be attributed to the recipient country. In line with this insight, exports are expected to reduce CO₂ emissions.

Moreover, imports encompass goods and services manufactured by a foreign country and consumed in another. The carbon emissions generated from goods and services must be accounted for domestically. It is projected that increasing exports will reduce CCO₂e in the host country, whereas increasing imports will increase CCO₂e in the host country. According to theory, a surge in imports is associated with an increase in consumption since imports constitute a significant component of any nation's aggregate consumption rate. As a result, imports from Uruguay account for a sizable portion of the host nations' intermediate and finished services and commodities.

There are two opposing views from a theoretical standpoint on the impact of financial development, especially in terms of environmental deterioration. First, financial development can help to improve environmental sustainability by dedicating more resources to clean energy, galvanizing the resources needed to invest in eco-friendly infrastructure, and assuring its viability in the long term [46]. Financial development also allows nations to adopt sophisticated technology for environmentally friendly and clean industries, thereby improving the sustainable environment [47]. Meanwhile, on the contrary, a greater level of financial development may have a negative impact on the environment. According to [2,11], financial development makes it simpler for corporations and individuals to gain access to small loans, allowing them to establish a new business or enhance an existing one. Hence, financial development enhances energy consumption, which has a negative influence on environmental quality. Since the theoretical viewpoint of the study is established, we proceed by constructing the study's functional form, which is as follows:

$$CCO_2e_t = f(EXP_t, IMP_t, REC_t, FD_t),\qquad(1)$$

where: CCO₂e, *EXP*, *IMP*, *REC*, and *FD* indicate consumption-based carbon emissions, imports, exports, renewable energy, and financial development, respectively.

### 3.2. Data

This study aims to investigate the influence of imports, exports, renewable energy, and financial development on CCO₂e in the case of Uruguay. We will achieve this by utilizing a quarterly dataset covering the period between 1990 and 2018. The information for CCO₂e is sourced from the Global Carbon Atlas, which is expressed in terms of carbon

emissions in million tons. The World Bank database is the source for export and import data, which are measured in the form of a share of GDP in percentage. Moreover, information on financial development is obtained from the International Monetary Fund database. Furthermore, the renewable energy data are obtained from the British Petroleum Database, and the metric for measuring renewable energy is kilowatts/per hour. The description of the variables is presented in Table 1.

**Table 1.** Description of variables.

| Variables | Metric | Sources |
|---|---|---|
| Consumption-based carbon emissions | Million tons | Global Carbon Atlas |
| Imports | Share of GDP in percentage | World Bank database |
| Exports | | |
| Renewable energy | Kilowatts/per hour | British Petroleum Database |
| Financial development | Index | International Monetary Fund database |

*3.3. Methodology*

3.3.1. Quantile Cointegration Test

The current paper assesses the cointegration between CCO₂ emissions and the regressors. We achieve this by employing the quantile cointegration test developed by [48]. By deconstructing the cointegrating procedure flaws into lead-lags, the traditional cointegrating methodology has endogeneity constraints that are solved using the ideas of [49]. The constant vector $\beta(\tau)$ is included in this model, which is an evolution of the cointegration model in [50] and is defined as

$$Y_t = \propto + \beta' Z_t + \sum_{j=-k}^{k} \Delta Z'_{t-j} \prod j + \mu_t, \tag{2}$$

$$Q_t^Y(Y_t | I_t^Y . I_t^z) = \propto (\tau) + \beta(\tau)' Z_t + \sum_{j=-k}^{k} \Delta Z'_{t-j} \prod j + F_u^{-1}(\tau) \tag{3}$$

The quadratic term incorporates the regressor and yields the following Equation:

$$Q_t^Y(\llbracket Y_t | I \rrbracket_t^Y . I_t^z) = \propto (\tau) + \beta(\tau)' Z_t + \gamma(\tau)' Z^2_{\ t} + \sum_{j=-k}^{k} \Delta Z'_{t-j} \prod j + \sum_{j=-k}^{k} \Delta Z^{2'}_{t-j} \prod j \ F_u^{-1}(\tau) \tag{4}$$

In Equation (4), the null hypothesis for the model is H0: $\beta(\tau) = \beta$ for all quantiles.

3.3.2. Quantile-on-Quantile (QQ) Approach

To examine how the variables of imports, exports, renewable energy, and financial development influence consumption-based carbon emissions at different quantiles, we deployed the [51] QQ estimator. The QQ approach is a more sophisticated version of the conventional quantile regression (QR) estimator, which is implemented when an investigator desires to discover how one parameter's quantiles impact the quantiles of another parameter. Therefore, the QQ approach integrates nonparametric and quantile regression techniques, in which one parameter's quantiles are regressed onto the quantile of another parameter.

The utilization of the QR approach in the applied analysis is classified into two phases. The conventional QR approach formulated by [52] covers the first phase, and [51] is used to evaluate the influence of regressors (hereafter, imports, exports, renewable energy, and financial development) on various quantiles of the endogenous parameters

(henceforth, consumption-based carbon emissions). The QR approach, as opposed to the conventional linear OLS estimator, is employed to investigate the influence of the regressors at both the center and tail of the endogenous parameter, allowing for a more extensive analysis of the associations between parameters. For the second stage, the local linear regression (LR) of [53,54] is used. The local LR assists in overcoming the "curse of dimensionality" issue that plagues nonparametric approaches exclusively. This dimensionality decrease approach involves establishing an LR locally surrounding each data point in the sample, with closer neighbors gaining higher weight. When these two modeling stages are combined, assessing the quantile connection between the endogenous variable and its regressors is crucial, generating more robust findings than other modeling techniques such as QR and OLS. In conclusion, it has been established that the QQ approach produces reliable findings when evaluating the effect of $X$ quantiles on $Y$ quantiles, which is used to assess the spatial effect of a single regressor on the endogenous variable. The QR approach is the foundation of the QQ approach, as expressed in the following Equation:

$$Y_t = \beta^{\vartheta}(X_t) + \mu_t^{\theta} \tag{5}$$

Here, the endogenous parameter is depicted by $Y_t$, $X_t$ represents the regressors, subscript $t$ represents the period of concern, the $\theta$th quantile of $Y$ conditional distribution is represented by $\theta$, and the error term of the $Y$ conditional distribution is depicted by $\mu$. Since the preceding information connecting $X$ and $Y$ is unidentified, the unknown function is represented by $\beta^{\vartheta}(.)$ This QR approach is utilized to estimate the impact of $X$ on the distribution of $Y$, which enables the influence of $X$ to fluctuate across several quantiles of $Y$. The fundamental advantage of this description is its stability, as there is no recognized hypothesis concerning the functional structure of the connection between $X$ and $Y$. Unfortunately, one shortcoming of the QR approach is its failure to detect total interdependence. In this regard, the QR approach neglects the possibility that the presence of $X$ disruptions impacts the interaction between $Y$ and $X$. In this connection, the outcomes of largely favorable $X$ disruptions, for example, can differ from those of relatively small, favorable $X$ disruptions.

Moreover, $Y$ will react asymmetrically to both favorable and unfavorable disruptions of $X$. As a result, $X^{\tau}$ neighborhood depicts the connection between the $\tau$th of $X$ and the $\theta$th quantile of $Y$; to assess the $X^{\tau}$ neighborhood, the local LR is deployed. The unknown function denoted by (.) can be approximated as a first-order Taylor extension centered on $X$ quantile, as shown below:

$$\beta^{\theta}(X_t) \approx \beta^{\theta}(X^t) + \beta^{\theta\iota}(X^t)(X^t - X^t) \tag{6}$$

Here, $\beta^{\theta}(X_t)$, the partial component about X, is shown by $\beta^{\theta\iota}$ and is also regarded as a response and is equivalent to the slope's coefficient in the framework of LR. The $(\beta^{\theta}(X^{\tau})$ and $\beta^{\theta\iota}(X^{\tau})$ parameters are indexed twice in $\tau$ and $\theta$, which is a significant characteristic of Equation (6). As $\beta^{\theta}(X^{\tau})$ and $\beta^{\theta\iota}(X^{\tau})$ are functions of $\theta$ and $X^t$ and $X^t$ is a function of $\tau$, it is evident that $\beta^{\theta\iota}(X^{\tau})$ and $\beta^{\theta}(X^{\tau})$ are both functions of $\tau$ and $\theta$, correspondingly. Furthermore, $\beta^{\theta}(X^{\tau})$ and $\beta^{\theta\iota}(X^{\tau})$ can be expressed as $\beta_0(\theta, \tau)$ and $\beta_1(\theta, \tau)$. As a result, Equation (6) can be rewritten as follows:

$$\beta^{\theta}(X_t) \approx \beta_0(\theta, \tau) + \beta_1(\theta, \tau)(X_t - X^{\tau}) \tag{7}$$

Once again, Equation (6) is substituted into Equation (5) and we obtain the following equation:

$$Y_t = \underbrace{\beta_0(\theta, \tau) + \beta_1(\theta, \tau)(X_t - X^{\tau})}_{(*)} + \mu_t^{\theta} \tag{8}$$

The $\theta$th conditional quantile Y is denoted by (*) in Equation (4). Meanwhile, unlike the conventional function of the conditional quantile, this display reveals the relationship between the $\theta$th of Y and the $\tau$th quantile of $X$ since the $\beta_0$ and $\beta_1$ parameters are indexed in $\theta$ and $\tau$ twice. These variables may change across different $\theta$th quantiles of $Y$

and $\theta$th quantiles of $X$. Moreover, no linear relationship between the variables' quantiles is assumed at any stage. Consequently, premised on the connection between their distinct distributions, Equation (5) measures the total dependence framework between $Y$ and $X$.

When estimating Equation (4), $X_t$ and $X^\tau$ are substituted by $\widehat{X_t}$ and $\widehat{X^\tau}$. The variables $b_0$ and $b_1$, which are $\beta_0$ and $\beta_1$ in the estimation, are acquired by resolving the following minimization concern:

$$\min_{b_0, b_1} \sum_{i=1}^{n} \rho\theta \left[ Y_t - b_0 - b_1(\widehat{X_t} - \widehat{X^\tau}) \right] K \left( \frac{F_n(\widehat{X_t}) \sqcap -\tau}{h} \right), \tag{9}$$

where $\rho_\theta(u)$ represents the function of quantile loss, $\rho_\theta(u) = u(\theta - I(u < 0))$, $I$ represents the indicator's regular function, and $K$ and $h$ depict the kernel function and the kernel bandwidth parameter, respectively. The Gaussian kernel is utilized to assess the observations' weight in the $X^\tau$ neighborhood, which is among the most commonly used kernel functions in economics and finance studies, owing to its computation, simplicity, and dependability.

At zero, the nature of the Gaussian kernel tends to be symmetric. It produces smaller weights in observations when they are farther away, thereby creating an indirect link to the distance between the analytical function's distribution of $\widehat{X_t}$, denoted by $F_n(\widehat{X_t}) = \frac{1}{n}\sum_{k=1}^{n} I(\hat{X}_k > \hat{X}_t)$, and the distribution value that correlates with the $X^\tau$, denoted by $\tau$.

Choosing the bandwidth is critical while deploying nonparametric techniques. Because it describes the neighborhood's scale, which encompasses the target position, the bandwidth affects the smoothness of the associated approximation. A larger bandwidth indicates a greater risk of estimate distortion, whereas a lower bandwidth indicates a greater risk of prediction uncertainty. Therefore, a bandwidth that offers a balance among variance must be determined. The study was conducted using a bandwidth $h = 0.05$ parameter, as recommended by [51].

The QQ approach is exceptionally suitable for this investigation because the theoretical connection between imports, exports, financial development, and renewable energy's effects on consumption-based carbon emissions is complicated, and modeling such a complex connection is burdensome with conventional linear econometric methods such as the OLS method. The QQ is also relevant in this investigation since the impact of imports, exports, renewable energy, and financial development on consumption-based carbon emissions is considered nonlinear and heterogeneous.

### 3.3.3. Robustness Test for the QQ Method

In this paper, we employ the quantile regression (QR) approach to examine the robustness of the QQ method's estimates. The variables of the quantile ($\theta$) are solely identified in this approach, which determines the $\theta$th quantile of regressors on CCO₂e. The coefficient slope of the QR technique, denoted by $\gamma_1(\theta)$ and described in the following Equation, estimates how trade openness influences GDP in a conditional distribution:

$$\gamma_1(\theta) \equiv \hat{\beta}_1 = \frac{1}{s} \sum_\tau \hat{\beta}_1(\theta, \tau), \tag{10}$$

in which the quantiles are denoted as $\tau$, which are 19 in numerical form.

### 3.3.4. Quantile Causality Approach

This research uses a unique technique suitable for accounting for nonlinear causation in quantiles, an invention of [55,56], which extends the framework in [57,58], where the endogenous parameter is denoted as $a_t$, whereas the regressors are denoted as $b_t$. According to [57], quantile-based causation may be defined as follows: considering the lag-vector, $b_t$ does not cause $a_t$ within the $\theta$ quantile of $\{a_{t-1}, \ldots, a_{t-p}, b_{t-1}, \ldots, b_{t-p}\}$ if

$$Q_\theta(a|a_{t-1}, \ldots, a_{t-p}, b_{t-1}, \ldots, b_{t-p},) = Q_\theta(a|a_{t-1}, \ldots, a_{t-p}). \tag{11}$$

When $b_t$ is $\{a_{t-1}, \ldots, a_{t-p}, b_{t-1}, \ldots, b_{t-p}\}$, $b_t$ does cause $a_t$ within the quantile ($\theta$), which is defined as

$$Q_\theta(a|a_{t-1}, \ldots, a_{t-p}, b_{t-1}, \ldots, b_{t-p},) \neq Q_\theta(a|a_{t-1}, \ldots, a_{t-p}). \tag{12}$$

Moreover, the $\theta$ quantile of $a_t$ is illustrated as $Q_\theta$ ($a \mid \cdot$). At is the conditional quantile, and Q ($a_t$) is based on $t$, with quantiles ranging from 0 to 1 (i.e., $0 < \theta < 1$). To recognize these vectors to demonstrate causation in quantile tests in a specific way, $A_{t-1} = (a_{t-1}, \ldots, y_{t-p})$, $B_{t-1} = (b_{t-1}, \ldots, b_{t-p})$, $C_t$ ($A_t$, $B_t$). $F_{a_t \mid C_{t-1}}(a_t|C_{t-1})$, and $F_{a_t \mid A_{t-1}}(a_t|A_{t-1})$ are the function of the mean, showing that the distribution functions of $y_t$ are based on the vectors $A_{t-1}$ and $C_{t-1}$ correspondingly. However, $F_{a_t \mid C_{t-1}}(a_t|C_{t-1})$) is the distributional mean, which is projected to predominate $a_t$ about every $C_{t-1}$. With the parameters $Q_\theta$ ($C_{t-1}$) $\equiv Q_\theta$ ($a_t|C_{t-1}$), and $Q_\theta$ ($A_{t-1}$) $\equiv Q_\theta(a_t|A_{t-1})$, the observation $F_{a_t \mid C_{t-1}}\{Q_\theta(C_{t-1} \mid C_{t-1}\} = \theta$ is correct when the probability is one. Bearing in mind the assumptions to evaluate for causality in quantiles based on Equations (13) and (14), which is expressed as

$$H_0 : P\{F_{a_t \mid C_{t-1}}\{Q_\theta(A_{t-1}|C_{t-1}\} = \theta\} = 1 , \tag{13}$$

$$H_1 : P\{F_{a_t \mid C_{t-1}}\{Q_\theta(A_{t-1}|C_{t-1}\} = \theta\} < 1 . \tag{14}$$

by using the distance metric [57], we evaluate the implementation of the causation within the quantile, which is represented as $J : \{\varepsilon_t E(\varepsilon_t|C_{t-1})f_Z(C_{t-1})\}$, in which the regression's error term is $\varepsilon_t$, and the marginal density function for $C_{t-1}$ is defined as $f_Z(C_{t-1})$. Therefore, the $\varepsilon_t$ impacts the quantile causation. The null hypothesis indicated in Equation (13), which happens when $E[1\{a_t \leq Q_\theta(A_{t-1}) \mid C_{t-1}\}] = \theta$, is used to compute $\varepsilon_t$. This statement has been modified to make $\varepsilon_t$ explicit. As a result, $1\{a_t \leq Q_\theta(A_{t-1}) = \theta + \varepsilon_t$, where the parameter's function is $1\{\cdot\}$. The distance metric based on Jeong et al. [57] is defined as the regression error, as shown in the following:

$$J = E\left[\left\{C_{y_t \mid c_{t-1}}\{Q_\theta(A_{t-1}) \mid C_{t-1}\} - \theta\right\}^2 f_C(C_{t-1})\right]. \tag{15}$$

Equation (14)'s metric distance conforms with $J \geq 0$, which is constrained by Equations (14) and (15). In terms of equality, the assertion is accurate, i.e., $J = 0$. However, this assertion is correct only if the null hypothesis is not invalid in Equation (13), whereas the alternative hypothesis is valid whenever $j < 0$ in Equation (14). Since the distance metric is defined in Equation (14), one of its potential analogs is the T-statistics of the quantile's causality that is kernel-based and has a predetermined quantile, as shown in the following:

$$\hat{J}_T = \frac{1}{T(T-1)h^{2p}} \sum_{t=p+1}^{T} \sum_{s=p+1, \, s\neq t}^{T} K\left(\frac{C_{t-1} - C_{s-1}}{h}\right) \hat{\varepsilon}_t \hat{\varepsilon}_s . \tag{16}$$

Note: $K(\cdot)$ and $h$ represent the kernel function and kernel approximation bandwidth for the specified vector $(C_t)$. $Th^p\hat{J}_T / \hat{\sigma}_0$ is distributed asymptotically as standard normal, according to [57], wherein $\hat{\sigma}_0 = \sqrt{2\theta(1-\theta)}\sqrt{1/(T(T-1)h^{2p})}$ $\sqrt{\sum_{t\neq s}K^2((C_{t-1} - C_{s-1})/h)}$. In T-statistics $(\hat{J}_T)$, the regression error is the most critical element. In this circumstance, the evaluation of the unknown regression error is

$$\hat{\varepsilon}_t = 1\{a_t \leq \hat{Q}_\theta(A_{t-1})\} - \theta , \tag{17}$$

Here, $\hat{Q}_\theta(A_{t-1})$ indicates the quantile estimator in Equation (16), which generates an estimate of the $\theta$th conditional $a_t$ given by $A$. The estimate $\hat{Q}_\theta(A_{t-1})$ is generated by employing the nonparametric kernel approach, which is shown in the following:

$$\hat{Q}_\theta(A_{t-1}) = \hat{F}^{-1}{}_{a_t \mid A_{t-1}}(\theta A_{t-1}).\tag{18}$$

Here, the Nadarya–Watson kernel estimator is indicated as $\hat{F}_{a_t \mid A_{t-1}}(a_t A_{t-1})$ such that

$$\hat{F}_{a_t \mid A_{t-1}}(a_t A_{t-1}) = \frac{\sum_{s=p+1,s\neq t}^{T} L\left(\frac{a_{t-1} - a_{s-1}}{h}\right) \mathbf{1}\{a_s \leq a_t\}}{\sum_{s=p+1,s\neq t}^{T} L\left(\frac{a_{t-1} - a_{s-1}}{h}\right)}\tag{19}$$

Hence, the function of the kernel is represented as $L(\cdot)$, whereas $h$ specifies the kernel estimate bandwidth employed.

The volatility's transmission is indicated by the causation of variance, which can emerge when there is no causality in the conditional distribution. The assessment of the Granger causality at the second moment contains specified issues, and it is critical to properly set up the procedure for this type of testing, considering that causality is rejected at the instant when *m* does not indicate non-causality at the period *k* for *m* < *k*. First, [58]'s Granger quantile causal method is employed. To show causality at higher periods, we first examine the following approach $a_t$:

$$a_t = \mathscr{g}(A_{t-1}) + \sigma(B_{t-1})\varepsilon_t\tag{20}$$

Here, the process denoted by $\varepsilon_t$ is dispersed identically and independently and the undetermined functions such as $\mathscr{g}(\cdot)$ and $\sigma(\cdot)$ satisfy the specific conditions necessary for stationary characteristics. While this formulation does not allow for nonlinear or linear causalities from $A_{t-1}$ to $a_t$, it allows $B_{t-1}$ to contain predicted information for $A_{t-1}$, given that $a_t{}^2$ is defined as the nonlinear function. Equation (21) illustrates that the nonlinear function does not always change with squares of $B_{t-1}$. As a result, in Equations (13) and (14), the H₀ (null hypothesis) and H₁ (alternative hypothesis) are re-modified within the variance of causality, as explained below:

$$\text{H}_0 : P\left\{F_{a_t^2 \mid C_{t-1}}\{Q_\theta(A_{t-1}|C_{t-1}\} = \theta\right\} = 1,\tag{21}$$

$$\text{H}_1 : P\left\{F_{a_t^2 \mid C_{t-1}}\{Q_\theta(A_{t-1}|C_{t-1}\} = \theta\right\} < 1.\tag{22}$$

Replace $a_t$ in both Equations (16) to (21) with $a_t^2$ to obtain a valid T-statistic for calculating H₀ in Equation (21). Considering that causality could exist in the variance (second instant), as well as causation in the conditional mean (first instant), there could be a problem with the causal interaction test based on the specification given in Equation (21), which is demonstrated in the subsequent model:

$$a_t = \mathscr{g}(B_{t-1}, A_{t-1}) + \varepsilon_t.\tag{23}$$

As a result, the causation in the second-order quantiles can be determined in the following hypotheses:

$$\text{H}_0 : P\left\{F_{a_t^k \mid C_{t-1}}\{Q_\theta(A_{t-1}|C_{t-1}\} = \theta\right\} = 1 \text{ for } k = 1, 2, \ldots, k;\tag{24}$$

$$\text{H}_1 : P\left\{F_{a_t^k \mid C_{t-1}}\{Q_\theta(A_{t-1}|C_{t-1}\} = \theta\right\} < 1 \text{ for } k = 1, 2, \ldots, k.\tag{25}$$

Along with the fundamental concept in play, to show how $b_t$ Granger keeps $a_t$ in the quantile ($\theta$) till the moment of *k*th is used in Equation (24) to generate Equation (17), which is the T-statistics of specific *k*, [58] use the density-weighted procedure to devise the Granger causality tests, comparable to [57], and show that in second moments, density-weighted nonparametric tests sustain the same asymptotic unconditional distribution as the causation test in the first instance. At the same time, other higher-moment specifi-

cations may be essential at times. Moreover, since the statistics are intertwined, it is impossible to analyze for all $k = 1, 2, ..., K$ simultaneously. To resolve this concern fundamentally, we use the sequential assessment approach proposed by [58] in checking for causality in Equations (21) and (24). At the first moment, we test for nonparametric Granger causality, i.e., $k = 1$, and then continue to evaluate for variance causality whenever non-causality is obvious. As a result, if $H_o$ is not rejected for $k = 1$, then the processes for $k = 2$ are subsequently built if there is causality in the second moment. This approach enables the existence of causality to be tested exclusively in variance, as well as the presence of causality in both the variance and mean at the same time. Finally, we may assess the existence of causality in both variances and mean across time. To use this method of causal assessment in quantiles, one must first determine the following three crucial options: the bandwidth $h$, the lag order $p$, and the kernel type for $K(\cdot)$ and $L(\cdot)$ in Equations (17) and (20), respectively. The lag order of 7 is used in this study to focus on the SIC inside the VAR about the determinant and $CCO_2e$. In addition, the SIC that can be used to overcome the problem of over-parameterization is more frugal than other criteria for picking delays. Least squares cross-validation procedures are used to find the bandwidth value. The $K(\cdot)$ and $L(\cdot)$ kernels are chosen using the Gaussian kernels.

## 4. Empirical Results

### 4.1. Preliminary Test Outcomes

The characteristics of all parameters will be discussed, summarized in Table 2, before the comprehensive empirical evaluation is carried out. As shown in Table 2, the average values of $CCO_2e$, EXP, IMP, REC, and FD are 2.1401, 3.1320, 3.1305, 3.8056, and 3.3590, respectively. However, the REC average value is the highest, whereas the lowest average value is $CCO_2e$ when compared to other parameters. In addition, $CCO_2e$ has the highest value in terms of standard deviation. The report for the skewness disclosed that all parameters are positively skewed except for $CCO_2e$ and EXP. All parameters are platykurtic in nature for kurtosis, except for FD, which is leptokurtic. Furthermore, at the significance level of 5%, the statistics of the Jarque–Bera confirmed that all parameters are not normally distributed given the rejection of the null hypothesis. Hence, since the data distribution of $CCO_2e$, EXP, IMP, REC, and FD is not normal, this serves as a justification for the use of the QQ approach. Finally, the stationarity properties of these series were ascertained using the KPSS and PP unit tests, as summarized in Table 3. The outcome of the tests indicates that all series are stationary at first difference.

**Table 2.** Descriptive statistics.

|  | **CCO₂e** | **EXP** | **IMP** | **REC** | **FD** |
|---|---|---|---|---|---|
| Mean | 2.1401 | 3.1320 | 3.1305 | 3.8056 | 3.3590 |
| Median | 2.1494 | 3.1557 | 3.0536 | 3.7722 | 3.2495 |
| Maximum | 2.6210 | 3.4811 | 3.5742 | 4.1070 | 4.2843 |
| Minimum | 1.5579 | 2.7073 | 2.8699 | 3.4935 | 2.9838 |
| Std. Dev. | 0.2911 | 0.2131 | 0.1909 | 0.16308 | 0.2977 |
| Skewness | −0.2492 | −0.2612 | 0.5439 | 0.5002 | 1.4911 |
| Kurtosis | 1.8907 | 1.9196 | 2.0495 | 2.3926 | 4.3987 |
| Jarque−Bera | 7.1480 | 6.9611 | 10.0872 | 6.6207 | 52.442 |
| Probability | 0.0280 | 0.0307 | 0.0064 | 0.0365 | 0.0000 |

**Table 3.** Unit roots outcome.

|  | **KPSS** | | **PP** | |
|---|---|---|---|---|
|  | **I(0)** | **I(1)** | **I(0)** | **I(1)** |
| CCO₂e | 0.0724 | 0.2463 * | −2.3141 | −4.8119 * |

| | | | | |
|---|---|---|---|---|
| EXP | 0.1237 | 1.5585 * | −2.1120 | −5.1189 * |
| IMP | 0.1957 | 0.7136 * | −0.7536 | −5.2604 * |
| REN | 0.2625 * | 0.1621 ** | −2.1371 | −5.5937 * |
| FD | 0.1266 | 0.2503 * | −2.1017 | −4.8810 * |

Note: * and ** represent 0.01 and 0.05 significance level.

### 4.2. Non-Linearity Test Outcomes

To further re-affirm the nonlinear properties of the parameters utilized, the current research deployed the BDS test, which is the innovation of [59], whose outcomes are summarized in Table 4. It suggests that all concerned series are nonlinear since the *p*-value for $CCO_2e$, EXP, IMP, REC, and FD is significant at the required level (i.e., between M2 and M6). Time-series data usually exhibit non-normal characteristics due to structural changes, financial crises, socioeconomic volatility, and a range of institutional and political issues [13,24]. As a result, the following estimation approaches are adopted premised on the non-linearity assumption. Given the properties of the data, the QQ approach is the best approach for concurrently taking non-linearity and structural modifications into account.

**Table 4.** BDS test.

| Dimension | $CCO_2e$ | EXP | IMP | REN | FD |
|---|---|---|---|---|---|
| M2 | 44.7617 * | 44.4099 * | 38.1529 * | 28.2532 * | 18.5641 * |
| M3 | 47.1428 * | 46.6071 * | 40.1524 * | 29.2406 * | 19.3796 * |
| M4 | 50.1467 * | 49.3602 * | 42.3827 * | 30.6910 * | 20.3522 * |
| M5 | 54.6470 * | 53.3648 * | 45.6022 * | 33.1269 * | 21.8835 * |
| M6 | 60.8736 * | 59.1472 * | 50.0714 * | 36.6792 * | 24.0877 * |

Note: * stands for a 1% level of significance.

### 4.3. Quantile Cointegration Outcomes

Table 5 shows the results of the quantile cointegration test, with the trade openness quantile in the τth symbolized by τ. The supremum norm values of coefficients (β and γ) reflect the stability of the observed parameters. As seen in Table 5, all of the coefficients of the determinants of $CCO_2e$ (β and γ) offer larger supremum norm values than all critical values, demonstrating evidence of a significant nonlinear or asymmetric long-run relationship between $CCO_2e$ and its determinants in Uruguay.

**Table 5.** Quantile cointegration test outcomes.

| | Coefficient | $Sup_\tau|V_\pi(\tau)|$ | CV1 | CV5 | CV10 |
|---|---|---|---|---|---|
| $CCO_{2et}$ vs. $IMP_t$ | β | 2211.36 | 1398.24 | 813.794 | 586.187 |
| | γ | 284.409 | 192.786 | 120.875 | 76.4328 |
| $CCO_{2et}$ vs. $EXP_t$ | β | 5628.83 | 3542.02 | 2333.68 | 1354.62 |
| | γ | 782.396 | 511.830 | 358.278 | 233.824 |
| $CCO_{2et}$ vs. $FD_t$ | β | 7054.79 | 5512.85 | 3683.32 | 1948.55 |
| | γ | 600.905 | 492.281 | 358.792 | 234.751 |
| $CCO_{2et}$ vs. $REC_t$ | β | 8471.78 | 6585.66 | 4661.35 | 2356.93 |
| | γ | 794.193 | 594.904 | 382.531 | 202.847 |

### 4.4. QQ Empirical Results

This section of the current study discusses the outcomes of the QQ technique regarding the impact of imports, exports, renewable energy, and financial development on consumption-based carbon emissions in Uruguay. The slope of the estimated coefficient $\beta_1(\theta, \tau)$ is illustrated in Figure 1a–d, presenting the influence of the quantile of $X$ ($\tau th$) on the

quantile of $Y$ ($\theta th$) at several values of $\tau$ and $\theta$ for Uruguay. The findings of the QQ approach are shown in Figure 1a–d. Figure 1a depicts the outcome of the QQ on the influence of exports on $CCO_2e$ in Uruguay. As shown in Figure 1a, the association between exports and $CCO_2e$ suggests a detrimental and strong influence on several quantiles, demonstrating exports' positive influence on environmental quality. In particular, a significantly strong and negative effect was detected in the region, which merges all quantiles of exports (i.e., 0.10–0.90) with the association across all quantiles of $CCO_2e$ (i.e., 0.1–0.95). Conversely, at the upper quantile of exports and mid-tail quantile of $CCO_2e$, the outcome indicates a weak and positive effect of exports in the mid-tail quantiles of $CCO_2e$. However, a substantial proportion of the quantiles suggest a negative relationship instead of a positive association. Thus, this outcome confirms that exports are an essential indicator that can help to achieve a sustainable environment.

In addition, Figure 1b reveals the outcome of the influence of imports on $CCO_2e$, which suggests the strong and positive effect of imports on $CCO_2e$ in a large proportion of quantiles. Specifically, the QQ approach detects a positive influence in regions that merge all quantiles of imports (i.e., 0.10–0.90) with the interconnection across all quantiles of $CCO_2e$. Moreover, the positive influence is relatively more robust in the lower quantile of imports and higher quantile of $CCO_2e$ than in the other quantiles. Concerning this outcome, we suggest that imports are a determinant that degrades the environment.

Furthermore, the estimates of the QQ approach for the influence of financial development on $CCO_2e$ are presented in Figure 1c. It suggests that a more significant fraction of the quantiles reveal a negative association, confirming the detrimental impact of financial development on $CCO_2e$. The degree of impact is relative across different quantile regions, as shown in Figure 1c. For instance, at the lower quantile of financial development (i.e., 0.1–0.3), in conjunction with the lower quantile of $CCO_2e$ (i.e., 0.1–0.4), as well as the middle and upper-tail quantiles of financial development (i.e., 0.4–0.95), in conjunction with the middle-tail quantile of $CCO_2e$ (i.e., 0.3–0.6), the degree of the detrimental effect is extreme compared to all quantiles. Thus, a negative association between financial development and $CCO_2e$ in Uruguay is evident.

Finally, the QQ approach's estimates of the influence of renewable energy on $CCO_2e$ are portrayed in Figure 1d. For many quantiles, a negative association can be confirmed between renewable energy and $CCO_2e$. In particular, the region that combines all quantiles of renewable energy (i.e., 0.10–0.90) had a considerably strong and negative influence on the relationship across all quantiles of $CCO_2e$ (i.e., 0.1–0.95). In contrast, at medium and higher quantiles of renewable energy (i.e., 0.3–0.9), the results show a strong and positive influence in the medium quantiles of $CCO_2e$ (i.e., 0.6–0.7).

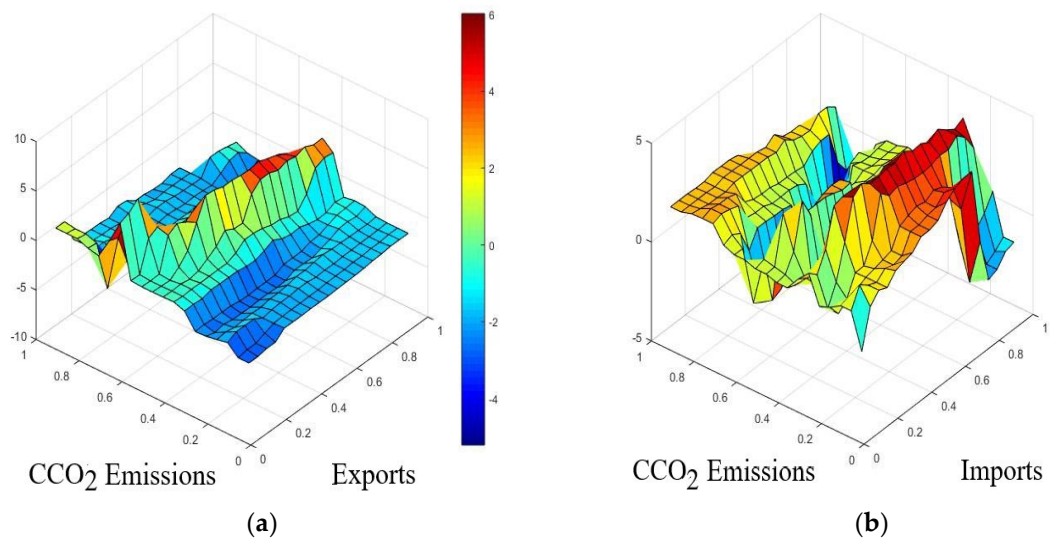

(a)                                                                 (b)

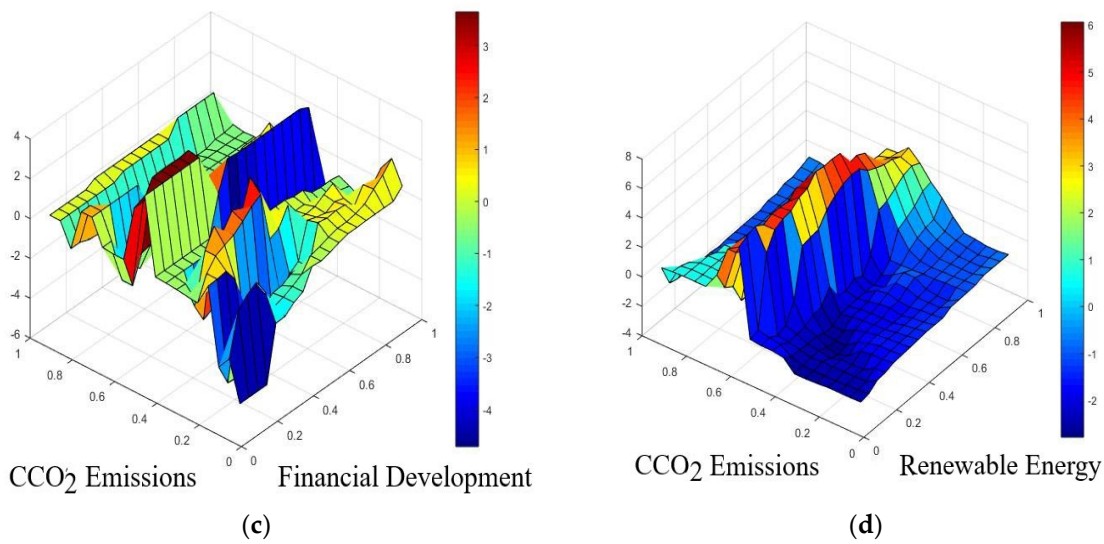

**Figure 1.** (**a**) Impact of EXP on CCO₂e; (**b**) impact of IMP on CCO₂e; (**c**) impact of FD on CCO₂e; (**d**) impact of REC on CCO₂e.

### 4.5. Robustness Check

In this section, the QQ approach estimates are compared with the estimates of the QR approach for Uruguay; meanwhile, the analysis of comparisons is presented in Figure 2a–d. Figure 2a–d shows that, regardless of quantiles, the averaged QQ estimates of the coefficient slope are extremely close to the QR estimates for China. This graphical presentation demonstrates that the properties of the QR model can be obtained by presenting the more detailed information included in the QQ estimates and a clear explanation of the QQ method. As a result, Figure 2a confirms the results of the QQ analysis described in Figure 1a. The QR results reveal that the influence of exports on CCO₂e is negative across all quantiles, which is consistent with the QQ regression result. Figure 2b shows that the QR outcomes are consistent with the QQ outcomes in Figure 1b, demonstrating that the influence of imports on CCO₂e is positive across all quantiles. In addition, Figure 2c confirms that the findings of QR reveal that the influence of renewable energy on CCO₂e in the majority of quantiles is negative, supporting the results of the QQ approach in Figure 1c. Finally, the influence of financial development on CCO₂e is negative, as depicted in Figure 2d, which corroborates the QQ approach outcome in Figure 1d.

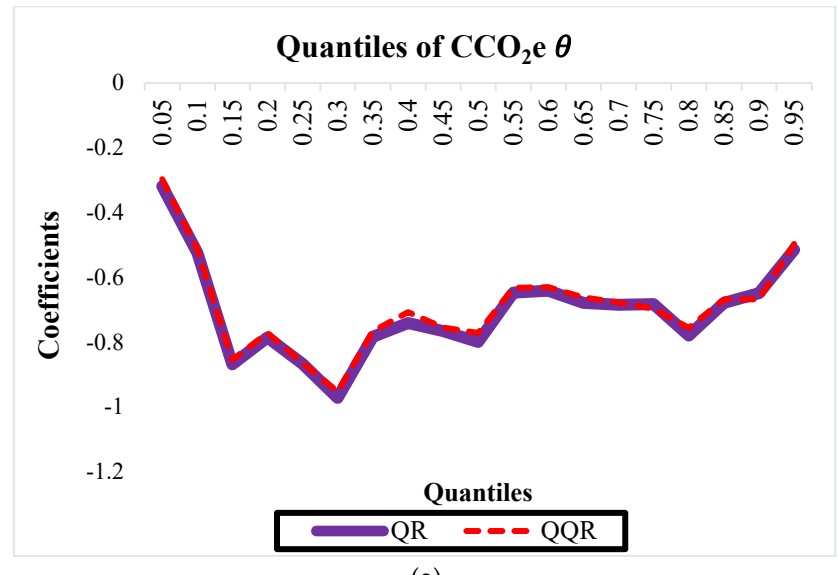

(**a**)

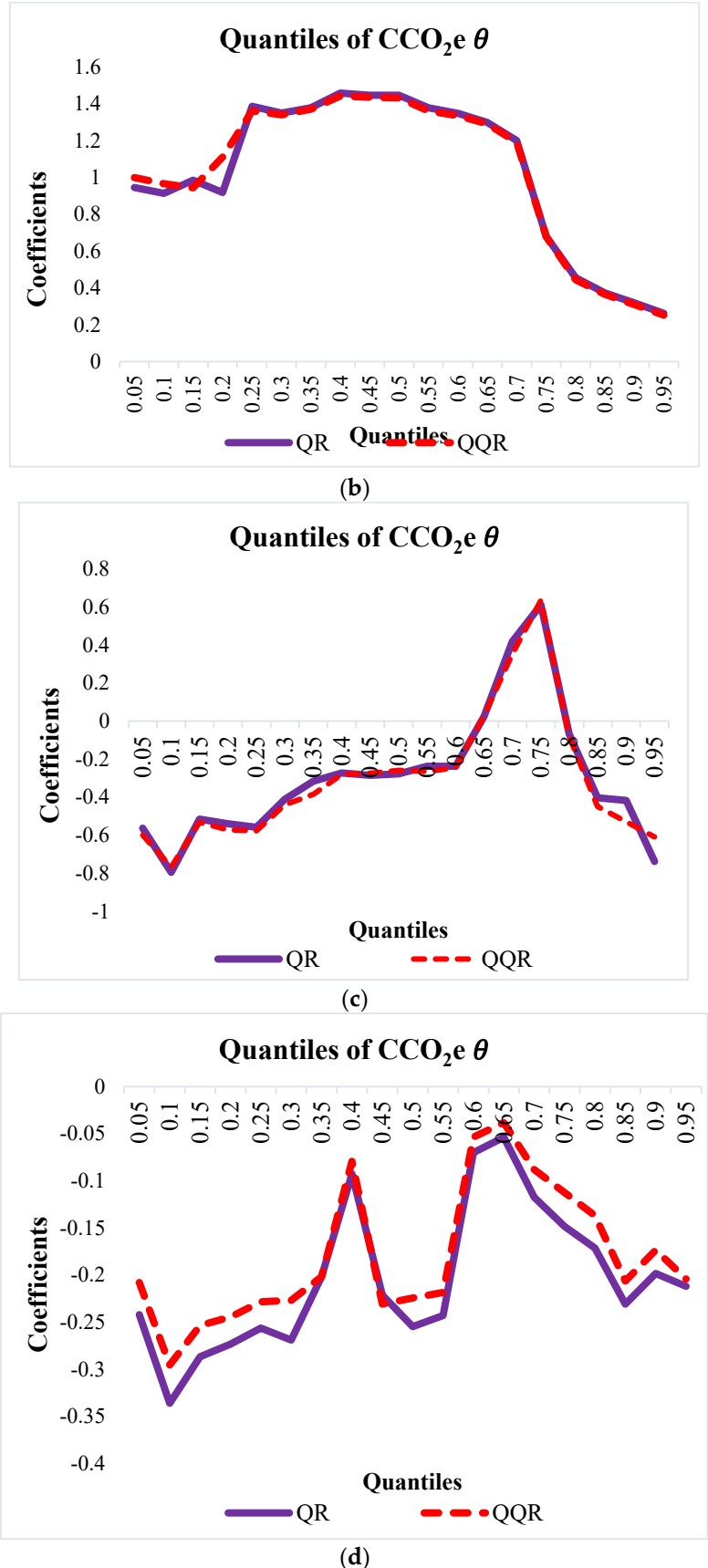

**Figure 2.** (**a**) Impact of exports on CCO₂e; (**b**) impact of imports on CCO₂e; (**c**) impact of renewable energy consumption on CCO₂e; (**d**) impact of financial development on CCO₂e.

*4.6. Granger Causality in Quantiles*

Since the influence of the regressors on CCO₂e has been considered, this study further employed the nonparametric Granger causality test in quantiles in both mean and variance, whose *p*-values are reported in Table 6 and Figure 3a–d. However, the quantile and T-statistics are represented in Figure 3a–d in the horizontal and vertical axis, respectively. Moreover, the green and dotted black lines signify the critical value of 90% and 95%, respectively. The red and purple lines denote the estimates of the *p*-values of the regressors for both volatility and mean, respectively.

In Table 6, we present the findings of the causal association from exports to CCO₂e. For export, causality is detected in the lower quantile (i.e., 0.15) at a 10% significance level. Moreover, at the middle-tail quantile (i.e., 0.3–0.6), we find evidence of causality from exports to CCO₂e. In addition, for the volatility of the exports, evidence of a causal interconnection from exports to CCO₂e is identified in the lower-tail (i.e., 0.10–0.15) and middle-tail quantiles (i.e., 0.25–0.5 excluding 0.4) at a 1% or 10% level of significance. The graphical presentation is provided in Figure 3a. Thus, the causal interconnection from exports to CCO₂e is asymmetric for volatility and a conditional distribution. Moreover, Figure 3b showcases the causal interconnection from imports to CCO₂e for mean and variance. For the mean distribution, the causality interaction from imports to CCO₂e is evident in the middle-tail quantile (i.e., 0.45–0.75) at a 1% or 10% level of significance, and for the volatility, the causal association is identified in the lower- and middle-tail quantiles (0.15–0.2 excluding 0.25) at a 1% or 10% level of significance. Therefore, an asymmetric causal interaction from imports to CCO₂e is evident for volatility and conditional distribution. Furthermore, Table 6 and Figure 3c verify that for the mean distribution, the causal interaction from financial development is present in the lower quantile (i.e., 0.15–0.2) and middle quantile (i.e., 0.5) at a 10% level of significance. Likewise, for volatility, causality is detected in the middle-tail quantile (i.e., 0.3–0.55) at a 1% or 10% significance level. Hence, we can conclude that there is an asymmetric causal interaction between financial development and CCO₂e. Lastly, the causal interconnection from renewable energy to CCO₂e for mean distribution is detected in the middle quantile (i.e., 0.45–0.55) at a 10% significance level. For the volatility, the causal connection from renewable energy to CCO₂e is confirmed in the lower quantile (i.e., 0.15) at a 10% level of significance, as well as in the middle-tail quantile (i.e., 0.30–0.60) at a 1% or 10% level of significance. These findings are graphically illustrated in Figure 3d. Thus, this finding concludes that there is an asymmetric causal interaction from renewable energy to CCO₂e in the volatility and conditional distribution.

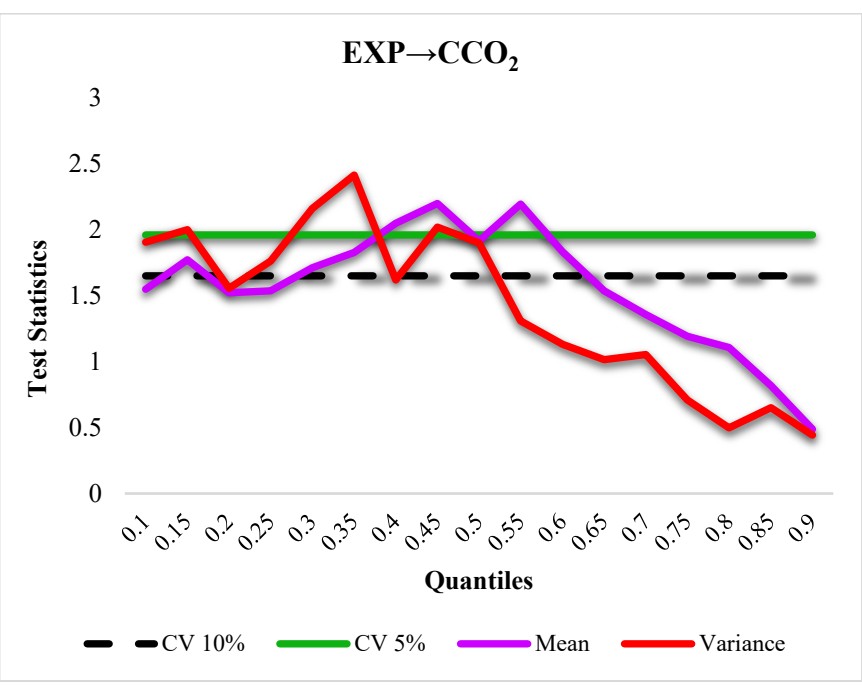

(**a**)

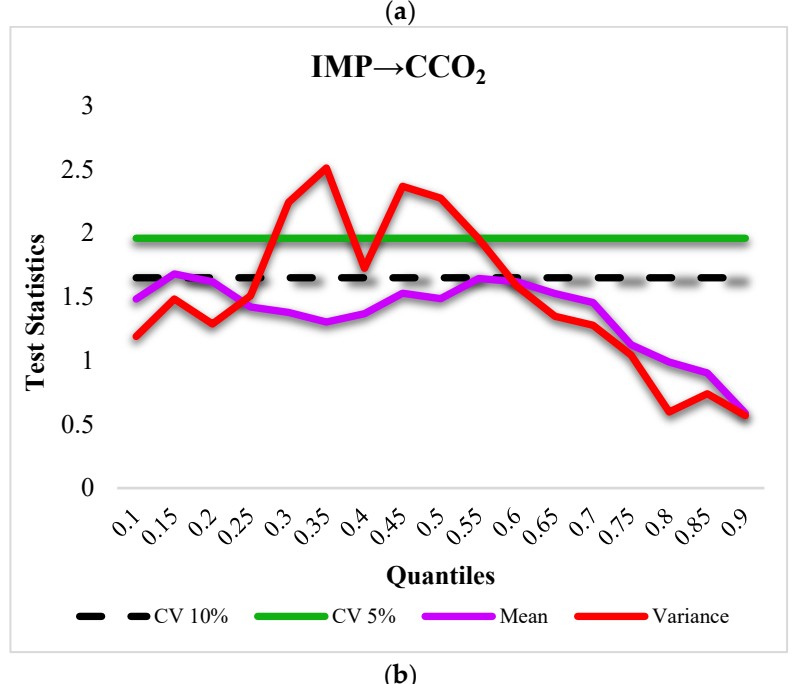

(**b**)

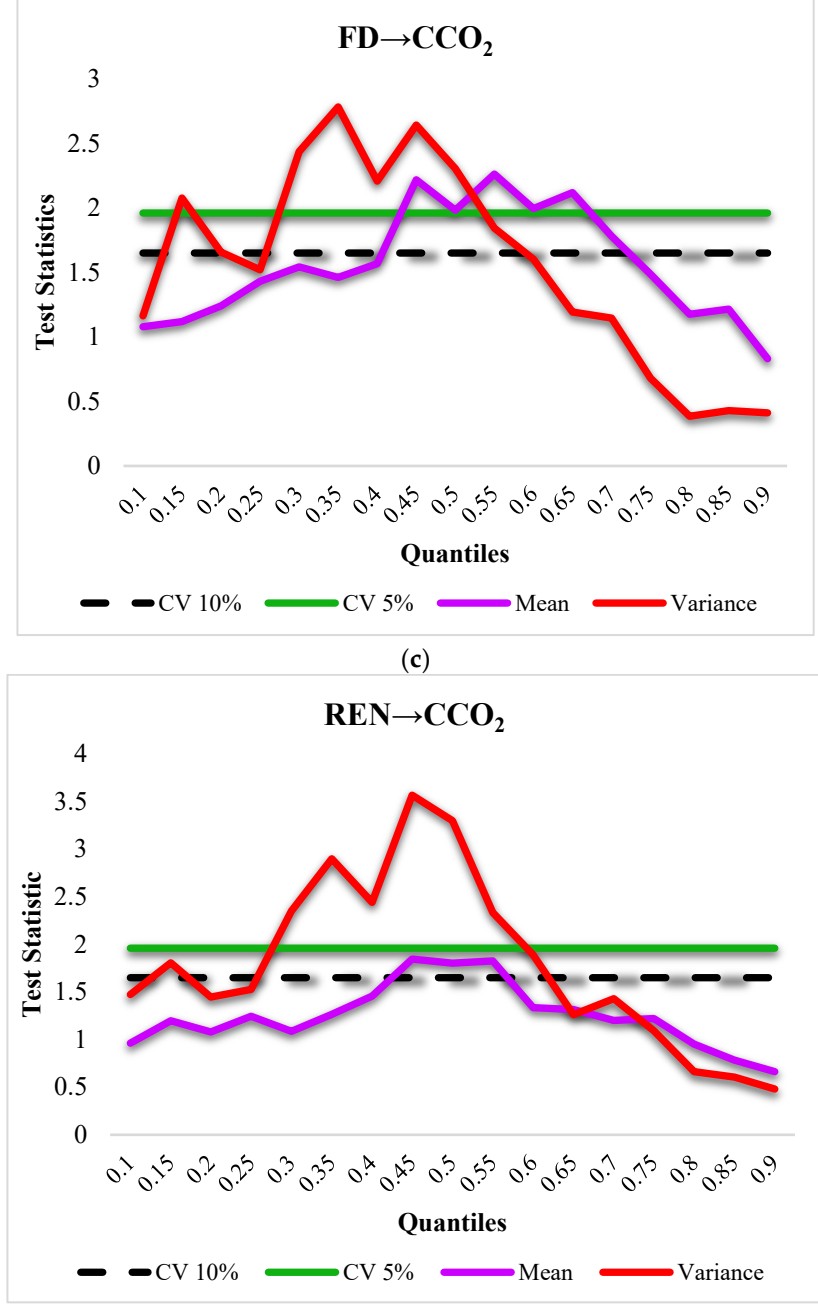

**Figure 3.** (**a**) Causality from EXP to $CCO_2$; (**b**) causality from IMP to $CCO_2$; (**c**) causality from REC to $CCO_2$; (**d**) causality from FD to $CCO_2$.

**Table 6.** Causality in mean and variance.

|  | CV-5% | CV-10% | Exports | | Imports | | Financial Development | | Renewable Energy | |
|---|---|---|---|---|---|---|---|---|---|---|
|  |  |  | Mean | Variance | Mean | Variance | Mean | Variance | Mean | Variance |
| 0.10 | 1.96 | 1.65 | 1.549 | 1.905 ** | 1.079 | 1.164 | 1.483 | 1.191 | 0.963 | 1.473 |
| 0.15 | 1.96 | 1.65 | 1.772 ** | 1.999 * | 1.119 | 2.076 * | 1.680 ** | 1.485 | 1.197 | 1.808 ** |
| 0.20 | 1.96 | 1.65 | 1.522 | 1.555 | 1.241 | 1.656 ** | 1.650 ** | 1.288 | 1.080 | 1.446 |
| 0.25 | 1.96 | 1.65 | 1.536 | 1.762 ** | 1.431 | 1.522 | 1.421 | 1.508 | 1.245 | 1.527 |
| 0.30 | 1.96 | 1.65 | 1.711 ** | 2.160 * | 1.542 | 2.436 * | 1.379 | 2.243 * | 1.086 | 2.352 * |
| 0.35 | 1.96 | 1.65 | 1.828 ** | 2.414 * | 1.461 | 2.782 * | 1.303 | 2.511 * | 1.266 | 2.901 * |

| 0.40 | 1.96 | 1.65 | 2.048 * | 1.618 | 1.568 | 2.207 * | 1.367 | 1.724 ** | 1.456 | 2.443 * |
| 0.45 | 1.96 | 1.65 | 2.198 * | 2.019 * | 2.219 * | 2.642 * | 1.528 | 2.367 * | 1.846 ** | 3.567 * |
| 0.50 | 1.96 | 1.65 | 1.915 ** | 1.901 ** | 1.982 * | 2.305 * | 1.485 | 2.275 * | 1.802 ** | 3.300 * |
| 0.55 | 1.96 | 1.65 | 2.193 * | 1.309 | 2.262 * | 1.843 ** | 1.654 ** | 1.956 ** | 1.825 ** | 2.334 * |
| 0.60 | 1.96 | 1.65 | 1.833 ** | 1.131 | 1.995 * | 1.602 | 1.620 | 1.583 | 1.337 | 1.890 ** |
| 0.65 | 1.96 | 1.65 | 1.536 | 1.014 | 2.118 * | 1.193 | 1.527 | 1.348 | 1.317 | 1.261 |
| 0.70 | 1.96 | 1.65 | 1.357 | 1.053 | 1.780 ** | 1.146 | 1.457 | 1.279 | 1.201 | 1.431 |
| 0.75 | 1.96 | 1.65 | 1.192 | 0.708 | 1.485 | 0.680 | 1.126 | 1.043 | 1.220 | 1.094 |
| 0.80 | 1.96 | 1.65 | 1.107 | 0.498 | 1.175 | 0.385 | 0.989 | 0.597 | 0.953 | 0.662 |
| 0.85 | 1.96 | 1.65 | 0.818 | 0.650 | 1.215 | 0.428 | 0.904 | 0.741 | 0.785 | 0.605 |
| 0.90 | 1.96 | 1.65 | 0.488 | 0.443 | 0.831 | 0.411 | 0.587 | 0.569 | 0.663 | 0.481 |

Note: * and ** denote $p < 5\%$ and $p < 10\%$, respectively.

### 4.7. Discussion

Following the presentation of the empirical analysis, the outcome needs to be discussed. For exports, the QQ approach findings reveal that exports' influence on $CCO_2e$ is negative, which was also corroborated by the QR approach. The finding confirms our theoretical conjecture that growth in exports offers additional services and goods for nations to consume while it leaves less to be consumed domestically, which serves as the justification for this outcome. This finding is supported by the studies of [22,25,33–35,60] that established a negative association between exports and $CCO_2e$; however, this result contradicts the findings of [61] for Turkey, which found that exports are positively associated with environmental degradation. Furthermore, this study also opposes the outcomes of [62] for 11 selected Asian economies, ref. [63] for seven selected Asian nations and [64] for 189 nations, which reported a positive association between exports and environmental degradation.

For imports, across all quantiles, the influence of imports on $CCO_2e$ is positive, according to the QQ approach. This finding is consistent with the estimates of the QR approach. This outcome is justified by our theoretical perspective, which concludes that growth in the degree of imports of goods and services is associated with greater consumption since it is seen as one of the essential factors in any nation's total level of consumption, especially when the nation is undergoing tremendous economic expansion at various phases, which is notably relevant in the scenario of Uruguay. Uruguay is a developing economy whose imports contain a considerable share of intermediate and final goods and services used by the host countries (Uruguay). This study's outcome complies with the study of [29] for China, ref. [31] for RCEP, and [26] for Italy, which reported a positive connection between imports and $CCO_2e$.

Furthermore, the findings of the QQ approach established that in all quantiles, renewable energy has a negative impact on $CCO_2e$, which was corroborated by the QR approach. This illustrates that an increase in REC improves environmental sustainability. The environment may be improved by renewable energy without hindering the process of progress, and it is also safe and sustainable. The most apparent reason for this interaction is that Uruguay substantially switched its energy mix from petroleum-driven energy generation to renewable sources between 2000 and 2019. Currently, renewable energy sources, primarily wind and hydropower, account for more than 98 percent of the total electricity generation in the nation. This outcome is in line with the studies of [40] for G7 economies, ref. [65] for MINT economies, ref. [26] for Italy, ref. [27] for Chile, and [28] for the Indian economy, which established the negative impact of renewable energy on $CCO_2e$. Furthermore, ref. [30] also complements our study by confirming the negative influence of renewable energy on $CCO_2e$ in Chile.

As revealed by the QQ approach and complemented by the QR approach, the effect of financial development on $CCO_2e$ is negative in Uruguay, demonstrating that the Uru-

guayan financial sector finances various environmentally relevant facets of economic activity. It suggests that Uruguay's financial development is moving in favor of sustainable practices and placing emphasis on environmental quality. Significant financial investment in environmental concerns helps to reduce emissions. Uruguay's financial sector is one of the most developed in Latin America. Uruguay is ranked 85th out of 183 nations in the Financial Development Index. As a result, it supports renewable energy projects, lowering the percentage of non-renewables in overall energy production and reducing reliance on imported non-renewables such as oil for energy production. This study's outcome is consistent with the research of [27], which concluded that financial development reduces the level of $CCO_2e$ in Chile. Furthermore, the studies of [36] for G8 nations, ref. [37] for OECD economies, and ref. [38] for selected Asian economies also corroborate our findings in the case of carbon emissions. Conversely, the results of [41] for 15 selected Asian economies, ref. [42] for UAE and [43] for South Asian economies contradict our study by establishing a positive association between financial development and environmental degradation.

Lastly, we used the nonparametric causality methodology pioneered by [55] to examine the causal influence of renewable energy, imports, exports, and financial development on $CCO_2e$ in Uruguay. The technique's benefit is its ability to analyze the variance and mean causal connection between variables. This result indicated that imports, exports, renewable energy, and financial development could contribute to significant variations in $CCO_2e$ at different quantiles. These findings have significant policy ramifications for Uruguayan policymakers.

## 5. Conclusions and Policy Recommendations

### 5.1. Conclusions

The current paper investigates the impacts of financial development, international trade, and renewable energy on $CCO_2$ emissions in Uruguay. In the authors' understanding, the influence of financial development, imports, exports, and renewable energy use on $CCO_2e$ in Uruguay has not been thoroughly investigated using our paper's newly established empirical methodologies. Thus, our paper bridges the literature gap for the case of Uruguay by employing the novel and newly developed quantile-on-quantile approach to investigate the impacts of the regressors on $CCO_2e$. Moreover, the quantile regression approach was used as a robustness test for the results of the quantile-on-quantile approach. In addition, the causal interaction from the regressors to $CCO_2e$ is evaluated using the nonparametric Granger causality test in quantiles. The dataset used for this analysis covers the period between 1990Q1 and 2018Q4. The plethora of findings from the empirical analysis is summarized as follows.

The findings of the QQ approach suggest that exports exert an adverse impact on $CCO_2e$. This is because exports have a negative association with $CCO_2e$ in the majority of the quantiles. The outcome implies that sustainable growth can be achieved in Uruguay through exports. The results of the QQ approach also indicate that imports could contribute to an environmental catastrophe. This is because the influence of imports on $CCO_2e$ is positive in all quantiles. Moreover, the QQ approach also disclosed that financial development mitigates environmental degradation since most quantiles disclosed that financial development's impact on $CCO_2e$ is negative. A similar outcome was discovered in the case of renewable energy and $CCO_2e$, in which the influence of renewable energy on $CCO_2e$ is negative in a large proportion of the quantiles. Furthermore, the quantile regression also authenticates the aforementioned outcomes. In addition, the outcome of the nonparametric Granger causality test in quantiles suggests that imports, exports, renewable energy, and financial development could cause significant variations in $CCO_2e$ at different quantiles.

### 5.2. Policy Recommendations

Based on the outcomes of our study, we recommend that emission-driven imports such as oil should be handled through non-restrictive trade measures whose sole target is minimizing carbon emissions, thereby mitigating the negative effect of imports. Since the import setup of Uruguay consists primarily of food and beverages, manufacturing equipment, and transportation, the country should prioritize importing environmentally friendly manufacturing equipment, which will not only mitigate the impact of imports on emissions but will also contribute significantly to reducing the externality impact induced by trade.

According to the present findings, Uruguay should utilize eco-friendly and cost-effective technology to facilitate a smooth transition to sustainable energy sources. Uruguay can reduce imports' adverse environmental effects (CCO$_2$ emissions) by implementing and investing in cleaner industrial technology. Uruguay should place a greater emphasis on technological innovation and shift the production sector from non-renewable energy consumption toward renewable energy use. This will assist the economy and the environment by lowering CO$_2$ emissions. Furthermore, the financial sector must strengthen its focus on giving funding to enterprises that embrace environmentally friendly technologies and incentivize them to employ other energy-efficient technologies for manufacturing reasons, thereby preventing environmental deterioration.

### 5.3. Limitations of Study and Extensions

Finally, we acknowledge the limitations of our research and acknowledge that it has the potential to be developed in the future. The current study used a bivariate analytical method, which may have been restrained due to the issue's complexity. To understand the causes of environmental deterioration, we adopted a baseline method. As a result, additional research can be conducted to investigate the impact of financial development, imports, exports, and renewable energy use on other determinants and other metrics of environmental degradation, including public-private energy investment, economic complexity, and institutional quality.

This paper employs the quantile-on-quantile approach to investigate the impacts of imports, exports, renewable energy, and financial development on consumption-based carbon emissions. Extensions of this paper include employing the quantile-on-quantile approach to investigate other issues—for example, profitability [66], volatility [67], market efficiency [68], and many others. This paper employs the quantile-on-quantile approach to investigate the relationship between energy and financial development. Another extension could use other approaches to study the relationship between energy [69–72] and financial development [73–75]. Readers may refer to [76,77] and many others for other topics to which academics and practitioners can extend the approaches used in our paper.

**Author Contributions:** Conceptualization, A.A.A. and T.S.A.; methodology, H.R.; software, T.S.A.; validation, T.S.A. and W.-K.W.; formal analysis, T.S.A.; investigation, A.A.A. and H.R.; resources, A.A.A.; data curation, H.R.; writing—original draft preparation, A.A.A. and T.S.A.; writing—review and editing, A.A.A. and W.-K.W.; visualization, A.A.A. and T.S.A.; supervision, W.-K.W.; project administration, A.A.A. and W.-K.W.; funding acquisition, W.-K.W. All authors have read and agreed to the published version of the manuscript.

**Funding:** This research has been supported by Near East University, Cyprus International University, Asia University, China Medical University Hospital, The Hang Seng University of Hong Kong, Hubei University of Economics, Research Grants Council (RGC) of Hong Kong (project numbers 12502814 and 12500915), and the Ministry of Science and Technology (MOST, Project Numbers 106-2410-H-468-002 and 107-2410-H-468-002-MY3), Taiwan.

**Acknowledgments:** The authors thank the anonymous referees for their helpful comments which help to improve our manuscript significantly. The fourth author would like to thank Professors Robert B. Miller and Howard E. Thompson for their continuous guidance and encouragement. However, any remaining errors are solely ours.

**Conflicts of Interest:** The authors declare no conflict of interest.

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
