# Peer review of "How Do Financial Development and Renewable Energy Affect Consumption-Based Carbon Emissions?"

_mca, doi:10.3390/mca27040073_

Round 1

Reviewer 1 Report

Title: How do Financial Development and Renewable Energy affect Consumption-Based Carbon Emissions in Uruguay?

Manuscript Number: mca-1703061

Journal: Mathematical and Computational Applications

It is my pleasure to review the manuscript for the esteemed journal. In this manuscript, the authors investigated the influence of import, export, renewable energy, and financial development on consumption-based carbon emission in Uruguay using quarterly dataset which covers between 1990 and 2018 and the novel quantile on quantile approach. The work presented is relevant to the Journal's field. The manuscript has got some potential. I would like to congratulate the author for a considerable amount of work that they have done. Especially, the authors reported that import, export, renewable energy, and financial development could envisage significant variations in consumption-based carbon emission at different quantiles. This manuscript has provided a new case to more comprehensive understanding the effects of import, export, renewable energy, and financial development on consumption-based carbon emission. However, the manuscript needs further improved before to be accepted for publication. The reviewer has listed some specific comments that might be helpful of the author to further enhance the quality of the manuscript. Please consider the particular comments listed below.

Comment 1: Abstract. It should further underscore the scientific value added of your paper in your abstract.

Comment 2: section of Introduction. As pointed out in the UNEP report, the COVID-19 pandemic has profoundly affected and changed the global and regional carbon emission. Therefore, the introduction section should not ignore the impact of the pandemic on the ecological environment. Please consider citing following paper: (i)https://doi.org/10.1016/j.envres.2021.111990; (ii) https://doi.org/10.1016/j.spc.2021.04.024  ; Added these citations will certainly improve the practical significance of the research of this article.

Comment 3: section of Introduction and literature review. The novelty of this revised manuscript should be further justified by highlighting main contributions to the existing literature. This could be clearly presented in the introduction section. Please consider citing following papers (i) renewable energy https://doi.org/10.1016/j.energy.2020.118200 ; https://doi.org/10.1016/j.jclepro.2022.130514 ; (ii) trade openness https://doi.org/10.1016/j.jclepro.2020.123838; https://doi.org/10.1016/j.techfore.2021.121465; On one hand, the current introduction seems simple. One the other hand, there has already been a large amount of literatures discussing this topic. There is a need to better elaborate the contribution of the work to the existing literature.

Comment 4: section of Theoretical Framework. The section is well-structured. However, it would be better to further highlight your improvement of the method and your innovation in methods.

Comment 5: sections of Empirical Results and Discussion. These two sections are also well-written and well-structured. However, it seems that there is no in-depth discussion. It would be better to further discuss what your findings are different from the past works.

Comment 6: section of conclusion and policy. Please make sure your conclusions' section underscore the scientific value added of your paper, and/or the applicability of your findings/results, as indicated previously. Basically, you should enhance your contributions, limitations, underscore the scientific value added of your paper, and/or the applicability of your findings/results and future study in this session.

Comment 7: There are still some occasional grammar errors through the manuscript especially the article ''the'', ''a'' and ''an'' is missing in many places, please make a spellchecking in addition to these minor issues.

Comment 8: References. Please check the references in the text and the list; You should update the reference. The line number is also missing.

Round 2

Reviewer 1 Report

  Authors have partially clarified my questions presented in my first report.

I have suggested some papers for improving this study, such as (i)https://doi.org/10.1016/j.envres.2021.111990; (ii) https://doi.org/10.1016/j.spc.2021.04.024; https://doi.org/10.1016/j.energy.2020.118200 ; https://doi.org/10.1016/j.jclepro.2022.130514 ; https://doi.org/10.1016/j.jclepro.2020.123838;https://doi.org/10.1016/j.techfore.2021.121465;

; but not all have been read and used, as a consequence the theoretical framework is still weak to provide a fruitful discussion. 

Limits of this study needs much more discussion and conclusions have not to be a summary of the paper. 

From this side, the paper remains limited.

Sure enough, a further revision of this paper not including suggestions provided in my reports to broaden the perspective of this study for the abovementioned issues is bound to be rejected.

Author Response

Thank you very much for your invaluable comments and suggestions, which have improved the revised version significantly.

We would also like to send our appreciation to you for your time and efforts in reviewing our paper.

We would also like to send our appreciation to you for your time and efforts in reviewing our paper and for providing excellent comments. Below are our responses to your helpful comments and suggestions.

Question 1. Moderate English changes required

Answer 1:  Thank you very much for your advice. We have polished our paper carefully.

Question 2. I have suggested some papers for improving this study, such as (i)https://doi.org/10.1016/j.envres.2021.111990; (ii) https://doi.org/10.1016/j.spc.2021.04.024; https://doi.org/10.1016/j.energy.2020.118200 ; https://doi.org/10.1016/j.jclepro.2022.130514 ; https://doi.org/10.1016/j.jclepro.2020.123838;https://doi.org/10.1016/j.techfore.2021.121465;

Answer 2:  Thank you very much for your advice. We have cited all the papers you suggested in our revised manuscript.

Question 3. Limits of this study needs much more discussion and conclusions have not to be a summary of the paper.

Answer 3:  Thank you very much for your advice. We have discussed some limits of our study and provided more discussion and conclusions to be a summary of the paper in our revised manuscript.

We hope that you will find this manuscript suitable to be included in an upcoming issue of your publication.

Reviewer 2 Report

Additional comments are as follows:

Comment 1: the data source needs to be clearly marked;

Comment 2: The format is not standard and needs to be adjusted again.

Comment 3: The name of Figure 1 should be placed at the bottom of the figure.

Comment 4: After the literature review, it is better to have a summary to better highlight the innovation.

Author Response

Thank you very much for your invaluable comments and suggestions, which have improved the revised version significantly.

We would also like to send our appreciation to you for your time and efforts in reviewing our paper. We would like to thank you for your following comments:

  • English language and style are fine/minor spell check required
  • Is the research design appropriate? (yes)
  • Are the methods adequately described? (yes)
  • Are the results clearly presented?  (yes)
  • Are the conclusions supported by the results? (yes)

We would also like to send our appreciation to you for your time and efforts in reviewing our paper and for providing excellent comments. Below are our responses to your helpful comments and suggestions.

Question 1. the data source needs to be clearly marked;

Answer 1:  Thank you very much for your advice. We have clearly marked the data source in our revised manuscript.

Question 2. The format is not standard and needs to be adjusted again.

Answer 2:  Thank you very much for your advice. We have adjusted the format of our paper in our revised manuscript.

Question 3. The name of Figure 1 should be placed at the bottom of the figure.

Answer 3:  Thank you very much for your advice. We have placed the name of Figure 1 at the bottom of the figure in our revised manuscript.

Question 4. After the literature review, it is better to have a summary to better highlight the innovation.

Answer 4:  Thank you very much for your pointing out the problem. We have made a summary to better highlight the innovation in our revised manuscript.

We hope that you will find this manuscript suitable to be included in an upcoming issue of your publication.

Round 3

Reviewer 1 Report

I appreciate the authors’ efforts made to revise their paper. This article has been improved. The manuscript is well organized and the problem addressed is quite interesting and clear. The objective and significance of the study are described soundly. My concerns from my previous review have been addressed. The authors have addressed most of the concerns I had with the previous review. Well, I would be favorable that this paper could be accepted for publicationThank you.

Reviewer 2 Report

The authors have incorporated comments from the first round of review. My concerns from my previous review have been addressed. I would recommend the paper to be accepted for publication.